# Supporting Depressed Mothers of Young Children with Intellectual Disability: Feasibility of an Integrated Parenting Intervention in a Low-Income Setting

**DOI:** 10.3390/children10060913

**Published:** 2023-05-23

**Authors:** Nasim Chaudhry, Rabia Sattar, Tayyeba Kiran, Ming Wai Wan, Mina Husain, Sobia Hidayatullah, Bushra Ali, Nadia Shafique, Zamir Suhag, Qamar Saeed, Shazia Maqbool, Nusrat Husain

**Affiliations:** 1Pakistan Institute of Living and Learning, Karachi 75600, Pakistan; 2Division of Psychology and Mental Health, University of Manchester, Manchester M13 9PL, UK; 3Department of Psychiatry, University of Toronto, Toronto, ON M5S IR8, Canada; 4Department of Psychology, Foundation University Islamabad, Rawalpindi 44000, Pakistan; 5TVI-Trust for Vaccines and Immunization, Head Office, Suite No 301, Al-Sehat Centre, Rafiqui Shaheed Road, Karachi 74000, Pakistan; 6School of Public Health, Dow University of Health Sciences DUHS, Karachi 74200, Pakistan; 7Department of Developmental-Behavioral Pediatrics, The Children’s Hospital, (UC HS-CH), University of Child Health Sciences, Lahore 54600, Pakistan; 8Mersey Care NHS Foundation Trust, Prescot L34 1PJ, UK

**Keywords:** depression, intellectual disability, parenting stress, parenting training, learning through play, feasibility trial, Pakistan

## Abstract

As a lifelong condition, intellectual disability (ID) remains a public health priority. Parents caring for children with ID experience serious challenges to their wellbeing, including depression, anxiety, stress and health-related quality of life. Integrated parenting interventions, which have been well evidenced for depressed mothers, may also effectively support depressed parents with a child with ID in low-resource settings such as Pakistan, and in turn optimise child outcomes. We conducted a mixed-method rater-blind feasibility randomised controlled trial, which assessed the feasibility and acceptability of the Learning Through Play in My Own Way Plus (LTP-IMOW Plus) intervention. Mothers who screened positive for depression (n = 26) with a young child (age 3–6 years) with ID were recruited from two low-resource community settings. Participants in the intervention arm (n = 13) received 12 group sessions of LTP-IMOW Plus and others (n = 13) received routine care. The intervention was feasible and acceptable with 100% retention and 100% session attendance. The intervention improved depression, anxiety, parenting stress and child socialisation score outcomes relative to the routine care arm. The framework utilised to analyse the qualitative interviews with seven participants at pre-intervention identified a range of struggles experienced by the mothers, and at post-intervention, found improved knowledge of child development and practices, improved mother–child relationships, recommendations for the intervention and perceived practical barriers and facilitators. The findings highlight the prospects for a clinical and cost-effective trial of an integrated parenting intervention to manage long-term parental mental health needs and improve child outcomes.

## 1. Introduction

Intellectual disability (ID) is a generalised neurodevelopmental disorder that places a heavy burden of disease on low and middle-income countries (LMICs) and has become a public health priority due to its nature of being a lifelong condition and its impact on families [1]. The global prevalence of ID is 3.2%; however, South Asia has the highest prevalence at 6.0% [2]. The incidence of ID in Pakistan is 1.1 per 100 live births [3]. Prevalence is particularly high in peri-urban slum areas due to poverty, malnutrition, birth trauma and consanguinity, which impact prenatal neurodevelopment [3]. The most common causes of ID, such as Down syndrome and fragile X syndrome, typically occur before birth [4]. The economic cost (family expenditure related to care and government expenditure on services provided for people with ID and their families) associated with ID is particularly high in severe ID [5], and this is a particular concern in LMICs where the financial impact falls on families [6]. In addition to the economic burden, families also experience social exclusion due to widespread stigma and cultural prejudices, such as perceptions of disability as pitiable and tragic [7].

Caring for a child with ID can be enormously challenging for family members, and mothers usually shoulder the most responsibility, which may explain their poorer wellbeing relative to fathers [8,9,10]. Parents of children with ID have high levels of depression and anxiety, sleep deprivation, poor quality of life, social isolation and stress [11,12,13,14,15,16]. In Pakistan, 70% of the parents of children with ID have anxiety and/or depression [17,18]. This may explain why children with ID tend to experience more unsupportive, negative parenting behaviours than typically developing children [19,20]. Comorbid child behaviour difficulties are associated with both parenting stress [12] and negative parenting behaviours [21]. Children with ID have also been found to experience less affectionate behaviours and fewer positive teaching behaviours from their mothers, such as joint attention and language [22]. On the other hand, a meta-analysis of fourteen studies suggested that positive parenting behaviours, such as praise/reward-giving and engagement/responsiveness were related to positive outcomes, such as social or adaptive behaviour, in young children with developmental disabilities [23].

In low-resource settings, the lack of specialised services further exacerbates the parenting stress as there are very few government-supported specialised education centres for children with ID in LMICs, and expensive privately-run centres are usually the only option [24]. Caregivers typically resort to informal care; however, home-based provision of care and education to children with ID is associated with depression and anxiety among their parents [25]. In Pakistan, available treatment or support for these parents follows an informal model of care as professional assistance is often scarce or unaffordable. Community-based treatment options (through lady health workers (LHWs), family-focused programmes, etc.) or informal sources of support are more readily available compared to formal support systems [17,26].

Parenting programmes are needed to improve the parenting skills, wellbeing and mental health among parents of children with ID [27,28]. In the early years, responsive parental behaviours—especially in mothers—are associated with enhanced cognitive and linguistic outcomes for children with ID [29]. More recent evidence in ID research has also highlighted the value of quality time in developing a close parent–child bond in the families of ID children [30]. Moreover, there is a growing evidence base for the salutary impact on parents who adopt meaning-focused coping strategies when dealing with their children who have ID [31]. In a systematic review, Beighton and Wills (2019) underpinned the experience of mothers in LMICs with having attained a wider perspective on life and their children living with the condition being a source of their contentment [31]. A similar finding was previously reported in Pakistani mothers despite the prevalence of cultural stigma and lack of social support [32].

Integrated parenting interventions with a focus on both parental mental health and development in typically developing children have shown a positive impact on a range of outcomes in low-resource contexts such as Pakistan [33,34,35]. To our knowledge, only one trial based in Pakistan has focused on the parents of children with neurodevelopmental disorders [36]. In a cluster randomised trial, a parent skills training module on developmental disorder was delivered to parents in a rural community in nine sessions by a family volunteer using a tablet-based Android application by adapting the World Health Organisation guidelines [36]. No difference was observed in child functioning, nor in parent–child joint engagement and the socioemotional wellbeing of children, but mothers’ health-related quality of life improved. Possible reasons for the former lack of effects include the heterogeneity of the developmental conditions, age range across the sample and the relatively brief duration of the intervention [36].

Learning Through Play in My Own Way (LTP-IMOW) is a parenting programme developed and tested for children with special needs and is based on learning through play (LTP) for typically developing children [37]. The LTP-IMOW component was integrated with existing culturally adapted cognitive behavioural therapy (CBT) to enhance maternal mental wellbeing [33,34,35]. The aims of this mixed-method study were: (1) to assess the feasibility and acceptability of group LTP-IMOW Plus CBT intervention in Karachi, Pakistan, and to test whether, relative to routine care, the scores on a range of outcomes (maternal health and parenting, child outcomes) moved in the direction of positive outcomes for the mother and child and (2) to explore the lived experiences of mothers having a child with ID and their experiences of participating in the intervention programme.

## 2. Materials and Methods

### 2.1. Design and Settings

This mixed-method study was conducted between March 2019 and October 2020. The quantitative aspect of the study adopted the rater-blind randomised controlled feasibility trial and the qualitative aspect followed descriptive qualitative design. The trial is registered on clinicaltrials.gov and the trial registration number is NCT03683706. The study was conducted in two low-resource community settings (Gadap Town; Orangi Town) in Karachi, Pakistan. These settings were chosen after consultation with stakeholders who were closely involved in LTP Plus studies in Karachi. This informed the decision to assess the feasibility of intervention in the settings with established community networks. Gadap Town has 8 Union Councils (UCs) and 400 villages, with around 15,000 births per year. Orangi town, the largest slum in Asia, has 13 UCs, and approximately 1.5 million people out of 15 million living in kachi abadi (unauthorised settlements on government land) in Pakistan [38]. Ethics approval was obtained from the Institutional Review Board of the Institute of Professional Psychology Bahria University Karachi (IPP/BU/OM/103/790) on 24 May 2016.

### 2.2. Recruitment and Randomisation

From each community, local community mobilisers, members from within a particular community, who can bring together as many stakeholders as possible to raise awareness of, and raise demand for a particular programme, to assist in the delivery of services, and to strengthen community participation for the sustainability of a particular programme. In this study, LHWs served the role of community mobilisers. These LHWs are part of the Lady Health Worker Programme that aims to promote health by bridging the gap between health services and community. Although these workers are attached to a health facility, they mainly work from home and are community-based, and have been referred by community leaders, identified potentially eligible mothers (during their routine visits to the community) for the study, and obtained consent to be contacted by the study team. Trained community health workers (CHWs) approached these women at their homes. Mothers were screened by CHWs using the study eligibility checklist. To be eligible, mothers were required to have at least one child with ID (including Down syndrome) aged 3 to 6 years living with her, as diagnosed by a health professional; a depression score ≥ 10 on the Patient Health Questionnaire (PHQ9) [39]; and the ability to participate in assessments and group intervention sessions, which could involve giving verbal responses to self-report measures. Mothers diagnosed with a severe physical or mental illness that would prevent study participation or with active suicidal ideation were excluded. CHWs provided eligible mothers with an information leaflet and CHWs described the study in detail. This participant information leaflet included contact details of free mental health helplines for participants. Following informed consent (thumb impression for those who were unable to read/write), baseline assessment was arranged with trained researchers at participants’ homes. Participants were assigned a study identification number by the project manager and were randomised into either the intervention arm (n = 13) (LTP Plus in My Own Way Plus CBT; ‘LTP-IMOW Plus’) or routine care arm (n = 13). Randomisation was carried out by an off-site statistician using a web software (http://www.randomisation.com, accessed on 19 July 2019). Within one week of randomisation, CHWs called participants and informed them of their randomisation status and scheduled a group session with participants in the intervention arm. Outcome assessments at the end of intervention (3-month post-randomisation) were completed by researchers blind to group allocation, and study participants were requested not to share their group status with the outcome assessors. The final sample size for this study (n = 26) was considered appropriate for a feasibility trial [40,41].

### 2.3. The Intervention: LTP in My Own Way Plus (LTP-IMOW Plus)

This is an integrated culturally adapted group parenting intervention comprising two components that run throughout the programme: a psychoeducational parenting intervention (LTP-IMOW) and CBT.

*LTP-IMOW* extends on the ‘Learning Through Play’ parent education programme developed by Toronto Public Health (1993) which was later revised by the Hincks-Dellcrest Centre (2000) [37]. LTP aims for the healthy growth and development of young children (birth to 6 years) using the stages and elements of development defined by Evans and Ilfeld (1982), incorporating Piaget’s theory of cognitive development [42] and Bowlby’s theory of attachment [43]. The objective of LTP is to strengthen parent–child attachment by encouraging parental involvement and by teaching parents how to use play activities to enhance their child’s development using the LTP calendar. The ‘Learning Through Play in My Own Way’ (LTP-IMOW) calendar enables the quick identification of areas and gaps in learning where detailed input by parents is required. This programme empowers parents with the knowledge that their child can be “different but equal”. It also enables the parents to become proactive partners in the planning and execution of the child’s developmental activities. Attractive pictures make the calendar user-friendly even for parents with low literacy. The LTP-IMOW calendar is different from the standard LTP calendar as the psychoeducational content was extended from five areas of child development (sense of self, physical development, relationships, understanding the world and communication) to seven areas, with physical development further divided into small muscles and large muscles and communication divided into understanding messages and expressing messages.

The CBT component focuses on psychoeducation with regard to maternal depression, the role of negative thinking patterns in the development and maintenance of depression, the impact of depression on child development, mothers’ daily functioning and their interpersonal relationships. In every session, there is a section on the basic principles of CBT to help mothers learn how to break the cycle of negative thoughts and to feel better [33,34,35].

An LTP-IMOW Plus intervention manual was developed by translating the LTP-IMOW manual into Urdu and integrating it with an existing culturally adapted CBT component for mothers of young children with probable depression [33,34,35]. In the LTP-IMOW calendar, culturally adapted illustrations were used to retain the cultural relevance of the intervention (such as pictures of parents and children with Asian physical appearance, clothing, environment/background picture, etc.). Both components of the intervention were delivered simultaneously, each session starting with content from the LTP-IMOW followed by the CBT component.

The group intervention was delivered by an LTP-IMOW Plus master trainer (masters level psychologist trained in LTP Plus with experience of delivering interventions in community settings in Pakistan) at the Basic Health Unit (BHU), in 12 weekly sessions over a three-month period. There were up to 7 mothers in each intervention group. Each session lasted between 60–90 minutes. Mothers were provided with a pictorial calendar, which they were advised to bring with them to each session.

### 2.4. Routine Care

Participants in the routine care arm received routine visits by LHWs, who are individually responsible for 150 households which they visit once a month. LHWs are trained in interpersonal communication and community engagement and cover all domains of maternal and child healthcare along with immunisation. For children with ID, usual care from primary healthcare centres in Pakistan consists of a range of dietary supplement regimens (multivitamin syrups and tablets) prescribed by primary health care physicians. In addition to usual care, participants received a phone call from the researcher, monthly for the first 3 months to ensure participants’ safety and engagement with the study. The research team maintained a record of usual care received by participants in both arms and none of them reported any psychiatric consultation, use of psychiatric medications or any psychosocial intervention.

### 2.5. Feasibility of the Study Procedures

The operational definition of feasibility in this study was addressed through the question: ‘can it work?’ [44,45]. The feasibility question was assessed by means of: the recruitment rate (the number of mother–child dyads referred by community mobilisers; the proportion of mothers who consented out of all eligible mothers who screened positive for depression referred from recruitment sites) and the attrition rate (the number of dyads withdrawn/number consented). The research team also evaluated the feasibility of administering the mother and child baseline and outcome assessment questionnaires (i.e., can existing validated Urdu-translated outcome measures be administered on this population?) and whether researchers can administer the assessment questionnaires appropriately. All researchers involved in this study maintained “research logs” to be evaluated by the study manager on a regular basis.

### 2.6. Acceptability of the Intervention

Acceptability operationalised as “the extent to which the participants receiving the intervention considered it to be appropriate” [46], was assessed through each participant’s “intervention attendance log”.

### 2.7. Qualitative Evaluation

Through a descriptive qualitative design, mothers’ experiences of depression and intervention participation were explored to gain insight into their lived experiences of having a child with ID and to assess the acceptability of the intervention. All 13 mothers in the intervention arm were approached to participate in a one-to-one qualitative interview at pre- and post-intervention stage; 7 mothers consented to participate. Two topic guides were developed based on the literature review and discussion with a multidisciplinary group of professionals which included mental health professionals, representatives from special education, and experts working in institutes offering services to children with special needs. The first draft of the topic guide was pilot tested with 1 participant in order to make sure that questions were understandable to the target population. The pre-intervention topic guide covered a range of areas: mothers’ knowledge and understanding of mental health problems (e.g., What are your views about mental health of parents caring for children with ID? What are factors contributing to maternal depression? etc.), the broad impacts of depression on children and family, the impact of having a baby with ID on a mother’s life, the knowledge of raising a child with ID and cultural beliefs and practices around ID. The post-intervention topic guide focused on exploring the perceived impact of LTP-IMOW Plus intervention, including mothers’ feedback on specific aspects of the intervention perceived as helpful (e.g., How did you find this training? What was helpful? What was not very helpful?), perceived changes in physical and mental health of mother and child, improvement in thinking style and problem-solving skills, any barriers and challenges experienced by the participants and suggestions for improving the intervention. All interviews were digitally recorded with consent and transcribed in Urdu. Relevant quotes were translated into English and then back-translated by independent bi-lingual researchers as a check.

### 2.8. Exploratory Evaluation of Outcomes

Maternal depression and child outcomes were assessed at baseline and on the third month post-randomisation for possible signals of effectiveness. All the assessment questionnaires except for the vineland adaptive behaviour scales were available in Urdu, and the Urdu versions have established psychometric properties. These assessment questionnaires have been used in previous trials on maternal depression in Pakistan [33,34,35].

#### 2.8.1. Maternal and Parenting Outcomes

The following well validated measures were mother-completed: the Patient Health Questionnaire (PHQ-9) [47], nine items rated on a 0–3 rating scale with a 0–27 range to measure the severity of depressive symptoms; the Generalised Anxiety Disorder (GAD-7) [48], a seven-item measure on a 0–3 rating scale with a 0–21 range of common symptoms of generalised anxiety disorder; and Euro-Qol-5 Dimensions (EQ-5D) [49], a 5-item generic, multi-instrument to measure health-related quality of life covering five dimensions of health (mobility, self-care, usual activities, pain/discomfort and anxiety/depression). We also utilised the Parenting Stress Index (PSI) [50], a 36 item measure covering three parenting stress domains related to parenting distress, difficult child characteristics and parent–child dysfunctional interaction, with items score on a 1–5 scale. 

#### 2.8.2. Child Outcomes

The Vineland Adaptive Behaviour Scales (VABS-III) [51]-parent/caregiver form were used, whereby mothers reported on the child’s adaptive behaviours with respect to communication, daily living skills, socialisation and motor skills, yielding an adaptive behaviour composite score. The measure includes 381 core items under a comprehensive section and 120 core items under a domain-level section, and the sections relevant for completion depended on the child’s chronological age, as per standardised procedure. 

### 2.9. Qualitative Analysis

Framework analysis was used to analyse the data [52] following steps recommended by Gale et al. [53]. (1) Transcription: Audio interviews were transcribed by trained researchers who were masters-level psychologists and was cross-checked against recordings by the qualitative researcher who conducted the interviews. (2) Familiarisation: During the familiarisation phase, the qualitative researchers actively read and re-read the transcripts to immerse themselves fully in the data. (3) Coding: Using an inductive coding approach, open coding was performed to generate as many codes as possible. Coding was performed using the Urdu transcripts by two qualitative researchers (TK, SH). A total of three transcripts from each dataset were coded by both researchers independently and then transcripts were divided between both researchers. (4) Development and application of analytical framework: To continue coding of the rest of the transcripts, a meeting was held with a qualitative research group composed of 5 senior qualitative researchers to agree on a set of codes to be applied to all subsequent transcripts. Codes were grouped together into categories (themes) and a working analytical framework. It was agreed that both researchers would apply the working analytical framework on the remaining transcripts and continue adding new codes under categories if needed. Both researchers had fortnightly meetings to discuss the codes from each transcript. Relevant codes were then translated into English for the purpose of analysis. To ensure the accuracy of translation, all were back-translated verbatim into Urdu by a bilingual researcher. (5) Charting: An MS Excel spreadsheet was used to organise the codes and categories using a matrix. The matrix included references to the illustrative verbatim from the dataset. This helped to visualise the data as a whole. (6) Interpretation of the data: Interpretation was informed by the study objectives and by the concepts generated inductively from the data. Rather than focusing on descriptions of individual cases, themes were developed to offer explanations for what was happening within the dataset.

To ensure the robustness of the qualitative findings, transcripts and themes were discussed in qualitative supervision meetings. Three transcripts from each of the two datasets (pre- and post-intervention) were independently analysed by two qualitative researchers and rest of the transcripts were coded by dividing transcripts between the qualitative researchers.

### 2.10. Statistical Analysis

Statistical Analysis was performed by using the Statistical Package for Social Sciences (SPSS) version 23.0 for windows. Independent samples *t*-test was calculated to evaluate the possible impact of the intervention on maternal, parenting and child outcomes at all distress levels (*p*-value 0.05). For scales that are divided into subscales (EQ-5D, PSI and VABS), *p*-value was corrected by dividing it with the total number of subscales for multiple testing within each questionnaire.

## 3. Results

### 3.1. Participant Characteristics

The mean age of participants in the intervention arm was 31 years, and routine care arm was 33 years. Most participants (77.0%) had no formal education. All mothers were home makers.

### 3.2. Feasibility Evaluation

#### 3.2.1. Recruitment and Retention

During 5 months of recruitment (from July to December 2019), CHWs successfully screened 35 participants from two recruiting centres. Of these, 26 (74.3%) mothers met the eligibility criteria and gave informed consent (Figure 1). None of them withdrew their consent; only one participant from the routine care arm had to leave the study as she relocated to another city, resulting in a retention rate of 96%. Participants were able to respond to all questionnaire assessments as evident from the completeness of data. Researcher logs suggested no issues with responding to the multiple choice measures.

#### 3.2.2. Acceptability of the Intervention

The attendance rate of the group intervention was 100%. All mothers attended all 12 intervention sessions. Travel facilities were provided to all mothers which facilitated them to attend all sessions. Moreover, there were no adverse events related to the intervention.

### 3.3. Qualitative Analysis

#### 3.3.1. Pre-Intervention Themes

In the pre-intervention interviews, six main themes were identified from the framework analysis (Table 1). The first theme focused on the mothers’ overwhelming sense of stress, burden, and impact on their mental health. Mothers reported a range of expressions for their depression, including the loss of pleasure, becoming upset, a deep feeling of hopelessness, crying, restlessness and sleep disturbance. Manifestation sometimes emerged through physical symptoms, such as headaches. The workload and burden described by most mothers led to irritability and anger, which were displaced on others, including their children with ID. Some experienced suicidal ideation and one described an attempt of self-harm due to the extreme distress they felt related to child disability and family conflicts. Some mothers also reported feeling upset and angered because of comments about the child’s apparent developmental delays made by wider family members and local neighbours who may have shown apparent concern that was perceived negatively by the mothers.

The second theme focuses on the mother’s concerns when she first observes an absence of communicative or other behavioural milestones that she expected would occur by a certain age. Some mothers initially observed that their child was not babbling and talking unlike other same-age children, did not ask for food, or was otherwise behaving differently. They reported motor delays, difficulty in verbal cues and understanding language, and differences in facial expressions. These social challenges or delays led to a sense of rejection for mothers, or a lack of rewarding experience. One mother expressed that “there is no bond between us because he does not understand my love and affection, he did not respond back to me when I try to express my love for him” (INT: 004).

Thirdly, increased childcare responsibilities were discussed. After the birth of the child, mothers reported that their responsibilities increased drastically. There is a strongly felt expectation that the complete responsibility of taking care of the child falls upon the mother. The mothers felt that they do not receive support from the wider family who needed help in supporting the child. On the contrary, mothers felt that they had to take care of other family members too. Most mothers reported being preoccupied with and constantly worrying about the child’s needs, which restricted all outdoor activities and was excessively draining.

Fourthly, some mothers relayed cultural beliefs in attempting to explain their child’s intellectual disability, specifically in relation to the role of the evil eye—a curse that can cause bad fate, ill health, or accident—that is documented in a range of cultures. This was discussed in the context of malevolent forces in the environment, perhaps linked to the negative feelings or intention of others outside of the family. These mothers reported taking precautionary actions on the basis of this belief (Table 1).

Finally, financial issues were raised by most of the participants as a barrier in accessing health care services and receiving treatment for their child’s intellectual disability. Some mothers reported that when they managed to visit a doctor for the first consultation, they were unable to manage follow-up visits because the treatment was too expensive for them.

#### 3.3.2. Post-Intervention Themes

In the post-intervention interviews, three main themes were identified with subthemes (Table 1).

The first theme focused on the perceived impact of intervention. Mothers endorsed benefits in different areas of their lives after participating in LTP-IMOW Plus group sessions.

Most mothers discussed how they had come to a different understanding about their child’s behaviour and difficulties and that this had altered how they were now ‘taking care’ of their child by utilising more responsive, positive and accepting parenting approaches. Mothers tended to describe why they changed their parenting behaviour in relation to the knowledge they had gained about child developmental limitations and delays. Participants also spoke in ways that suggested a new focus of empathy for their child’s mental states, demonstrated by how they had started spending more time with their child, talking and playing. This contrasts with pre-intervention interviews when the focus was on the mother’s own mental states. Through expressing a newfound or refreshed awareness of the affected child’s needs for relationships, a healthy diet, language interaction, etc., most mothers implicitly recognised their pivotal role in helping to meet those needs. In this context, some mothers described how they were able to observe improvements in their child, for example, in their receptive language.

In a second subtheme, all mothers reported improvement in their attachment with the child and expressed a strong awareness of the importance of meeting the child’s emotional needs. Most of the mothers previously assumed that, due to their child’s condition, they could not develop a bond with the child as the child was not able to understand feelings. Now, as they started to pay more attention to their child, they have observed that the child can express feelings of joy and, by responding to this joy, the mother feels more connected to the child. Therefore, a positive feedback loop is described in which parenting has become a more positive experience for mothers. All mothers also acknowledged that they learned how the child feels more secure when parents provide them with appropriate attention and responsiveness, and some mothers described perceptible effective changes in the child, with reduced distress and more positivity.

In a third subtheme, most mothers expressed a noticeable improvement in their own sense of mental wellbeing. A heightened sense of self-efficacy was described as they learned to identify their unhealthy negative thoughts and the ability to change these thoughts. Some participants also mentioned that they learned problem-solving skills and the significance of self-care, and how these directly and indirectly impacted on their family relationships.

The second theme focused on suggestions made by the participants for further refinement in the intervention. Most participants suggested that this training should be delivered to their partners as well so that together they can bring positive change in their child through improved parenting styles. They also suggested increasing the timing of the sessions to be able to discuss their concerns in more detail.

Thirdly, mothers discussed barriers and facilitators they experienced in attending the group sessions. Barriers centred around time constraints due to the burden of household activities and childcare responsibilities. However, all mothers successfully managed the perceived barriers and attended training sessions. One facilitator mentioned: support from their family members helped them participate in the group sessions, such as scheduling a group session.

### 3.4. Exploratory Outcomes

As a feasibility study, the study was not designed (i.e., powered) to test formal hypotheses. However, *t*-tests can provide indications of the intervention’s potential effectiveness. First, the degree of outcome variable overlap was checked. Based on correlations between all outcome variables, all PSI subscales were highly correlated with the total score (baseline: r = 0.64 to 0.86; all *p* < 0.01; follow-up: r = 0.93 to 0.98; all *p* < 0.01), so only the PSI total is reported. VABS domains varied in how much they were inter-correlated (baseline: r = 0.33; *p* > 0.05 to r = 0.74; *p* < 0.001; follow-up: r = 0.42; *p* > 0.05 to r = 0.81; *p* < 0.001); see Appendix A), and given that they are typically reported separately as distinct developmental areas, each VABS domain was reported. Overall, there were few correlations between the nine outcome variables at baseline, and many more at follow-up (see Appendix A). The adjustments were not made for multiple comparisons given the exploratory nature of the study.

#### 3.4.1. Maternal and Parenting Outcomes

Maternal depression (PHQ-9) and anxiety (GAD-7) scores did not differ significantly between groups at baseline, but were significantly lower in the intervention arm at follow-up (Table 2). Total parenting stress (PSI) scores showed no difference between groups at baseline, but had significantly reduced at follow-up for the intervention arm.

Health-related quality of life did not differ between groups at baseline, but was significantly higher for the intervention group than the routine care group at follow-up, based on both the EQ-5D descriptive score and visual analogue scale (self-rated health) score (Table 2).

#### 3.4.2. Child Outcomes

VABS socialisation did not differ between groups at baseline, but there was a highly significant increase in the intervention arm relative to routine care (Table 1). Communication, daily living, socialisation and motor skills increased for the intervention arm from baseline to follow-up assessment. *p*-values indicate that differences between two groups lack statistical significance (except for the socialisation subscale) (Table 2). No significant differences were found by intervention arm at baseline or follow-up in VABS communication, VABS daily living or VABS motor skill. Child communication skills were substantially lower in the routine care group than the intervention arm, while daily living scores changed slightly for both groups.

## 4. Discussion

This mixed-method study aimed to assess the feasibility and acceptability of a culturally adapted group parenting intervention for mothers who had screened positive for depression and whose children were diagnosed with ID in Pakistan: LTP-IMOW Plus. The study demonstrates excellent recruitment, uptake and retention of mothers who screened positive for depression. Unlike high-income settings where formal support for families is available, in low-resource settings such as Pakistan, parents receive minimal support from formal sources due to the limited availability of trained support staff and poor organisational structures [17]. A lack of support infrastructure may explain the excellent recruitment and retention rates in this study. Qualitative findings highlight that mothers who participated managed to attend group sessions, despite barriers related to work overload and time constraints. Moreover, facilitation in terms of travel-related arrangements for participants may have contributed to engagement with the intervention as evidenced from previous qualitative work with South Asian populations [54].

Qualitative interviews captured the burden felt and severe mental health impacts associated with parenting a child with ID, and a shift in both the parenting experience and their relationship with their child after participating in the intervention programme. Thus, both strands of data support the potential benefits of the integrated LTP-IMOW Plus intervention in improving parental depression, anxiety, stress and a health-related quality of life, and child outcomes.

The quantitative findings on group differences should be interpreted cautiously as the study was not powered for formal hypothesis testing. However, the findings are promising and informative for designing the next comprehensive RCT, e.g., when analysing the requested sample size for the RCT trial. Future investigation should determine the clinical and cost-effectiveness of LTP-IMOW Plus, and may also investigate the engagement of depressed fathers of children with ID.

While early interventions for vulnerable children can bring improvements to familial wellbeing and children’s development [55], the key challenges to LMIC settings are their availability and accessibility to families who stand to benefit the most [27]. Pre-intervention themes in this study highlight financial difficulties as barriers in accessing services and treatment in Pakistan. Psychoeducational support is essential to equip families with the knowledge to care, train and support their children with ID and enable them to learn self-care strategies [56]. Consistent with other low-resource environments, families’ cultural beliefs may impact how they raise children with ID [56,57], as highlighted through our qualitative finding that (some) participants associated ID with supernatural causes and beliefs.

The quantitative and qualitative findings regarding the intervention’s potential effectiveness on parent, parenting and child outcomes are preliminary, yet are consistent with previous studies using a similar LTP intervention with depressed parents of typically developing young children in Pakistan [33,34,35,58]. Parenting interventions delivered to mothers of children with ID in high- and low-income settings indicated that positive parenting had a significant effect on improving somatic symptoms, anxiety, depression, social dysfunction, and the overall mental health of mothers [27,28,59,60]. Findings on the potential benefits for health-related quality of life are particularly important as parents of children with ID experience lower quality of life due to behavioural problems in children, leading to increased psychological distress [61].

Existing evidence has established the need for research on the role of a culturally sensitive parenting programme and potentially efficient group-based parent education programmes for the families of children with intellectual disabilities [25]. The apparent changes in mothers’ knowledge and attitude towards child development and parenting described in the mothers’ interviews, combined with indicative findings in child outcomes, can only be confirmed in a full trial, as has been evidenced in other LTP Plus trial works in typically developing young children in Pakistan [35].

This study offers several strengths: it used a mixed-method design to triangulate findings, a feasibility study for assessing the practicalities of randomising this particular population in a trial, and key indications of effectiveness in maternal and child outcomes were evaluated. Several limitations must also be acknowledged. As participants were recruited through community networks, results may differ if recruiting from settings such as healthcare facilities, education, or rehabilitation settings. Eligibility was based on having a child with a confirmed diagnosis of ID by a health professional; no information was collated on the level of intellectual disability or the presence of comorbid diagnosis. Mothers were screened for depression using a validated screening instrument (PHQ-9); no diagnostic instrument confirmed a diagnosis of a major depressive episode. Acceptability, based on attendance, would be improved by assessing the level of engagement. The test of feasibility was based on intervention and outcome assessment completion without any issues; a formal checklist determined a priori would have provided more nuanced information for informing areas for modification or refinement. Although outcome measures were compared between the two groups (intervention vs. routine care), mean differences should be interpreted cautiously; this feasibility study was not powered to formally test for effectiveness. Furthermore, routine care did not involve any formalised contact with services, so group differences could be due to the intervention itself or more generally from group support. The long-term engagement of the mothers of children with ID in a trial is not known from this study as the follow-up assessment was at 3 months post-randomisation. The routine care group was not interviewed about their lived experiences of caring for a child with ID or of being part of usual care, both of which may have been informative for taking the intervention work forward.

## 5. Conclusions

To our knowledge, this is the first study which assessed the feasibility and acceptability of an integrated parenting intervention for mothers who screened positive for depression and whose children were diagnosed with ID in Pakistan. This culturally adapted, manual-assisted, integrated parenting intervention was well-tolerated by mothers with probable depression of young children with ID. Establishing the feasibility and acceptability of potentially low-cost, easy-to-deliver parenting interventions is important in laying the foundation for larger clinical and cost-effective trials in LMICs, where awareness and resources regarding mental health in general and ID in particular are limited. Moreover, the findings of this study highlights the opportunities to develop feasible, acceptable, potentially cost-effective, and community-level interventions, which can be up-scaled and integrated into existing healthcare systems. Community-based interventions ensure maximum population coverage, with an emphasis on managing problems that are prioritised by the end beneficiaries, such as improved parental mental health and improved child adaptive behaviour.

## Figures and Tables

**Figure 1 children-10-00913-f001:**
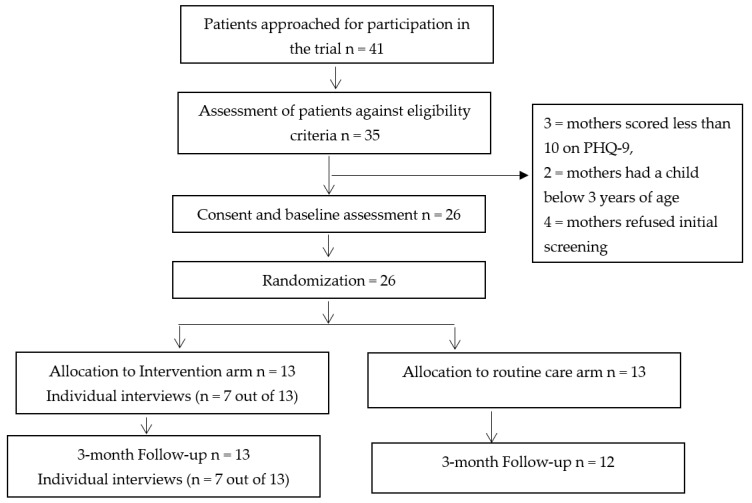
Consolidated standards of reporting trials—CONSORT flow diagram.

**Table 1 children-10-00913-t001:** Participants’ verbatim quotes from pre- and post-intervention interviews.

	Pre-Intervention Themes	Post-Intervention Themes
1	Maternal stress, burden, and mental health: “Depression is a feeling of deep hopelessness and sorrow. You just want to cry and after crying you feel a bit lighter but again you are in the same state. I feel tired all the time just like I am dead inside” (INT:002). “I am not feeling well mentally, I am upset all the time and my headache is never relieved” (INT: 004). “I am really overburdened and it makes me really irritated and aggressive. I displace my anger on my children. I get aggressive and I beat my children when I hear anything against my disabled child, I often beat my other children and ask them to leave home. I am really worried” (INT: 006). “My health is getting affected, I feel very angry and think I should die, I get annoyed easily as I have to take care of him all the time” (INT:003). “People do not understand that the child has limited abilities and they interact with the child in a harsh manner. Nobody believes that she is a disabled child. Sometimes I beat her when she does not understand things. All the time she gets beaten by one or another family member, except her father, all others beat her because she does not behave like normal children” (INT:001).	1: Perceived impact of intervention 1.1: Improved knowledge of child development and practices: “After attending training, I got to know how important it is to talk to the child. Now I talk to him while working like during cooking and let him know names of different things, now he can recognise a few objects. Whatever work I do, I leave if my child needs me. I got to know that I need to take care of him, play with him, make him sit with other children so that he can build relationships” (INT 001). “Until 6 months ago I did massage my child then I stopped but now we know that massage increases the development of the child. I observed that after giving a massage he looks fresh, and he can sleep well. We should also take care of the child’s diet as my child is not able to express hunger needs like other normal children” (INT 003). “Sometimes he is happy, sometimes he cries but because he cannot speak so he cannot express his distress. Previously I just let him cry but now I try to know the reason for his crying as he cannot tell himself” (INT 005).
2	Concerns related to child development: “I recognised the problem after 5 months of her birth. I observed that she never babbles, a normal child starts babbling and looking at others after 40 days of birth. She cannot speak, she cannot speak a complete word till this age. We noticed that she never asks for food” (INT:002). “I thought he was weak because I did not eat well during pregnancy. When I consulted a doctor I asked why he was so weak. Other children start sitting at the age of 6 months. Why he is not able to do so, he cannot sit, cannot change his posture, so he (the doctor) said that he is mentally disabled, he is different from other children, he will start talking at the age of 5 to 6 years” (INT: 005).	1.2: Improved mother–child attachment: “Now I play with my child, and I observe that he feels happy. I spend more time with my child, and I feel that he is getting attached with me, previously I did not feel connected with him. He comes to me and gives me a smile; I feel so happy” (INT 001). “The way you teach us to take care of child health not only physically, but also their emotional needs. It was a new thing for me as I never thought to take care of my child this way. We just focussed on feeding him, ignoring their emotional needs, as to us he is not smart like other children” (INT 004). I keep him with me so that he can understand that I am his mother and he can feel secure. I used to leave him alone and stayed busy with other tasks and he used to cry all the time but now even if I am busy I keep him with me and now he does not cry very often” (INT 005).
3	Increased childcare responsibilities: “People think the child is the responsibility of the mother only, but it’s not. Taking care of a disabled child is a huge responsibility that you cannot take alone. You always need support from your husband and family, but people don’t understand rather they ask you to do things for them as well” (INT:001). “I cannot go anywhere as I am always worried about her food, and sleep. Nobody can take care of her except me, and also I have other household tasks to do. I feel really drained by the end of the day” (INT: 007).	1.3: Perceived improvement in maternal wellbeing: “To some extent now I have control over my thoughts, and I am trying to change my thinking style also. Previously I used to talk aggressively but I learnt that I should not talk aggressively to anyone because it will impact my relationships and my mind as well” (INT 001). “We should patiently think what the solution of the problem is. We should try to stay happy most of the time. I was so preoccupied with my child that I forgot to take care of myself. In this programme, I get to know we should also take care of our own self, if we take care of our own self only then we would be able to take care of our family, specifically our children” (INT 003).
4	Cultural beliefs regarding child intellectual disability: “We don’t take him outside after evening prayer or at night because it’s the time of evils and they attack children” (INT 006). “He was absolutely fine after birth but after a few months we observed changes, maybe we live in a village and there are a lot of trees near our house so this (ID) could be due to evil forces. Sometimes I think this is due to an evil eye, maybe people were jealous of my pregnancy”(INT 001).	Suggestions to improve the parenting intervention: “It should also be arranged for fathers; they should also get to know about the child’s needs. Moreover, I felt that an hour and half is short. It should be more, as we are in a group session, so everyone has a lot of things to talk about and discuss” (INT 007).
5	Financial issues as a barrier in receiving treatment for a child: “His treatment is so expensive, and we cannot afford it. So, I don’t give him medicines and cannot take him to the doctors” (INT: 003). “We can’t get the treatment; we are poor people we can’t spend money on follow up visits to doctors” (INT: 004). “We have very few resources, we live in a village and my husband does not earn a handsome amount. We hardly arrange money to get food, so it’s difficult to manage a child needs and treatment. We are hopeless and can’t get the treatment” (INT: 004).	Perceived barriers and facilitators: “The only barrier was household responsibility and leaving the child at home, but I tried to manage this” (INT 002). “I did not have any issue; you were coming to our village to train us. Initially it was difficult to manage time but then I managed time accordingly” (INT 002).

**Table 2 children-10-00913-t002:** Comparison between two groups on maternal and child outcomes.

Outcomes	Mean (SD) Routine Care	Mean (SD) LTP-IMOW Plus	Mean Difference (95% CI)	*t*	*p*-Value
**PHQ**					
**Baseline**	15.31 (2.057)	17.00 (1.780)	−1.692 (−3.251 to −0.134)	−1.708	0.113
**FU**	18.23 (6.044)	4.67 (4.141)	13.564 (9.240 to 17.888)	9.782	<0.001
**GAD-7**					
**Baseline**	13.54 (1.050)	13.23 (1.423)	0.308 (−0.705 to 1.320)	0.627	0.536
**FU**	16.15 (5.257)	4.67 (4.559)	11.487 (7.400 to 15.575)	5.813	<0.001
**PSI total**					
**Baseline**	149.31 (25.41)	149.23 (26.160)	0.077 (−20.801 to 20.955)	0.008	0.994
**FU**	160.54 (28.690)	76.08 (30.007)	84.455 (60.168 to 108.742)	7.194	<0.001
**EQ-5D**					
**Baseline**	0.43 (0.260)	0.52 (0.257)	−0.096 (−0.305 to 0.114)	−0.943	0.355
**FU**	0.45 (0.314)	0.79 (0.266)	−0.337 (−0.578 to −0.095)	−2.880	0.008
**EQ-5D VAS**					
**Baseline**	46.54 (8.987)	49.23 (6.405)	−2.692 (−9.010 to 3.625)	−0.880	0.388
**FU**	48.08 (8.549)	63.33 (7.785)	−15.256 (−22.041 to −8.472)	−4.652	<0.001
**VABS Communication**					
**Baseline**	14.85 (15.842)	22.23 (15.303)	−7.385 (−19.993 to 5.224)	−1.209	0.239
**FU**	20.23 (25.437)	33.83 (18.155)	−13.603 (−32.031 to 4.826)	−1.527	0.140
**VABS**					
**Living Skills**					
**Baseline**	9.62 (7.848)	8.92 (6.538)	0.692 (−5.155 to 6.539)	0.244	0.809
**FU**	8.62 (7.880)	9.00 (6.368)	−0.385 (−6.344 to 5.575)	−0.134	0.895
**VABS**					
**Socialisation**					
**Baseline**	15.31 (16.038)	18.00 (14.289)	−2.692 (−14.988 to 9.603)	−0.452	0.655
**FU**	12.08 (13.895)	25.42 (11.958)	−13.340 (−24.109 to −2.570)	−2.562	0.017
**VABS**					
**Motor Skills**					
**Baseline**	20.08 (16.307)	24.69 (25.866)	−4.615 (−22.119 to 12.888)	−0.544	0.591
**FU**	26.46 (22.980)	26.08 (18.579)	0.378 (−17.005 to 17.761)	0.045	0.964

PHQ-9: Patient health questionnaire–9. FU: Follow-up. GAD-7: Generalised anxiety disorder. PSI: Parenting stress index. PD: Parenting distress. P-CDI: Parent–child dysfunctional interaction DC: Difficult child. EQ-5D: EuroQol-5 Dimensions. VAS: Visual analogue scale. VABS: Vineland adaptive behaviour scales.

## Data Availability

Requests for sharing the anonymised data should be addressed to the lead author.

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
