# Peer review of "Supporting Depressed Mothers of Young Children with Intellectual Disability: Feasibility of an Integrated Parenting Intervention in a Low-Income Setting"

_children, 2023, doi:10.3390/children10060913_

Round 1
Reviewer 1 Report
Please see the attached file.
I also add the comments here:
First of all: thank you for a very well written and important paper and the work that is done for these mothers! However, there are some points with which you can imporve the paper.
1. Background:
The background section is structured and provides a brief, but comprehensive overview of the current research available, a clearly defined definition and the contextualization of intellectual disability and the burden such a diagnosis has on society and the family itself. The author then provides more information on the impact on maternal mental health, and current interventions were discussed. The intervention LTP-IMOW and the research project's aims are clearly defined. The following suggestions should be taken into consideration:
· Even though the gap within the body of knowledge is highlighted, it would be helpful to provide the reader with other positive parenting studies and their outcomes.
· Current routine care within Pakistan should also be discussed
2. Method:
The methods section consists of 9 subsections. The following was of note within each section:
· Design and Settings:
o The research design was mentioned. Furthermore, the ethical considerations were adequately discussed. A brief contextualization of the target group was discussed.
o It would be helpful if the author highlighted which method was used for the qualitative and quantitative research
· Recruitment and Randomization
o The recruitment process, as well as the inclusion and exclusion criteria, was clearly defined. Ethical considerations were made. The authors also discussed the randomization process sufficiently.
o It would be useful if the author explained how the sample size was calculated.
· The intervention: LTP in my own way plus
o The intervention aims and the theory it is grounded in are discussed. The two components of this intervention are briefly mentioned.
o It is unclear if there are any contra-indicators for this intervention. Furthermore, it would be beneficial for the reader to clarify whether the two components occur simultaneously or in a specific order. Lastly, the author can consider elaborating on the qualifications of the master trainer.
· Routine care
o The usual care is discussed; however, it is recommended that the researcher elaborates on the resources and care available.
o One ethical consideration – The target group is vulnerable. Were the participants in this group able to also receive appropriate mental health support after the intervention?
· Feasibility of the study procedures & Acceptability of the intervention
o Both these sections were adequately discussed.
· Exploratory evaluation of outcomes
o Maternal depression and child outcomes were discussed, and assessments were also mentioned. Measurement moments were also mentioned:
o A title was missing in section 2.7.1
o It would be helpful for the reader to know if these assessments were standardized for this population group
· Qualitative evaluation
o The researcher discussed pre- and post-intervention interviews. There were a few significant suggestions for this section.
o It is unclear why only people from the intervention group were considered for these interviews.
o Furthermore, it would be helpful to understand how the topic guide was developed and what the general questions asked.
o Lastly, the authors should consider, for replication purposes, defining and discussing the qualitative method used. It would also be helpful to understand how the reliability and validity of the themes identified were determined.
· Statistical Analysis
o This section could be elaborated upon as it only gives an overview of the methods used.
3. Results:
This section is clear and concise, with relevant results. The participant characteristics were discussed briefly, and the author discussed the feasibility evaluation of the study. The figure within this section was precise. The quantitative outcomes were discussed, and the table is informative and provides enough information.
· Even though the demographic information is discussed in the table, it may be helpful to provide a summary in the text of each assessment used.
The Qualitative analysis starts with Table 2. It would be useful if the author provided a brief introduction to give the reader an overview of the sub-section and how these themes were chosen. The results of the themes were discussed adequately. It would be helpful if the author referred to table 2 and its quotes. Lastly, it is recommended that the author states how many interviewees mentioned the theme being discussed.
4. Discussion:
The discussion section integrated and briefly summarised this study's aims and quantitative and qualitative results. Furthermore, the authors could link it back to the greater body of knowledge within this field of research. However, it would be recommended that the author mention the themes and sub-themes identified in the qualitative results. A new methodological aim is discussed in this section. It is recommended that the researcher integrate this into the original aims mentioned in the introduction. Future research recommendations were made. However, the limitations of the research project need to be defined more.
5. Conclusion:
The conclusion briefly studies the research project and the results obtained. This is clearly done.

Reviewer 2 Report
Kindly refer to the attached document. Thank you.

Reviewer 3 Report
Dear Authors
The subject is very intresting, your methodology and results were well presented. You also mentioned the limits of your study and the main objective. I would also have liked to know what are the prospects for the future concerning the implementation of this method in the course of care for these children, who do not have the possibility of following medical care.
Reviewer 4 Report
Thank you for the opportunity to review the present paper on “Supporting depressed mothers of young children with intellectual disability (ID): Feasibility of an integrated parenting intervention in a low income setting”. The paper describes a feasibility study of the program “Learning through play in my own ay plus (LTP-IMOW Plus)” in low resource settings in Pakistan. After careful recruitment 26 mothers with depressive symptoms participated in the study and were randomized either to the intervention group (n = 13) or to the routine care group (n = 13). The authors investigated feasibility and acceptability of the program. In addition, they conducted qualitative interviews to obtain an in-depth understanding of key themes of the mothers at pre-intervention and at 3-months follow-up. Quantitative data were also assessed at both time points.
In my view, this paper is clearly of interest to the readers of CHILDREN. It is very well written, profoundly designed and the authors are to be applauded for conducting such a comprehensive study in low-resource community settings in Pakistan. The topic of ID is also highly relevant and interventions that address parental symptoms are essential.
Notwithstanding these positive aspects of the study, there are some major concerns and some minor points that I would like to address.
Major concerns:
My main concern relates to the statistical analysis of the quantitative data, see p. 4, line 181. The authors frame this topic as “exploratory evaluation of outcomes”. They assess maternal depressive symptoms, stress and anxiety symptoms at baseline and at 3-months follow-up. In the description of the statistical analysis, see p. 5, line 218, they describe paired t-tests as the statistic but refrain from reporting significance values. In Table 1, however, t-values are not reported but mean differences and 95% CI are presented. To me, this is confusing and I would recommend to clarify and rewrite this section entirely. First, you should follow a sound statistical procedure to analyze the data, i.e. decide on the statistical analyses (e.g., paired t-tests), describe all the statistical key parameters, including p-values, be aware of multiple comparisons and present all these findings in a Table. In my view, you can state that these analyses are exploratory in nature, given that they do not form your primary research aim. However, all statistical information is necessary for the reader to understand your findings.
Table 1 should therefore be improved. In addition, I would suggest highlighting the significant results, as I got confused given that some comparisons were not statistically significant.
This profound information can then form the basis for the next comprehensive RCT, e.g. when analyzing the requested sample size for the RCT trial.
In line with my comment, I suggest restructuring the method and result section so that your main research questions – feasibility, acceptability, qualitative data – are presented first. The quantitative data can be presented after that.
Another major concern relates to the sample of mothers with depressive symptoms. I suggest rewording the description of the mothers throughout the paper. If I understand the procedure correctly, mothers were categorized as “depressed” based on scores on the PHQ9 (p. 3, line 106). This score is indicative of depressive symptoms but does not allow a diagnosis of a depressive disorder.
Minor comments
I would suggest adding the term “feasibility study” to the title and the abstract.
P. 3, line 110: Could you comment on what happened to mothers with suicidal ideation?
P. 4: CBT: Given that mothers were not diagnosed with a full-blown depressive disorder how did you inform them about their scores on the questionnaire?
P. 4, line 181: How many mothers were able to fill in the questionnaires by themselves?
P. 5, line 199: Could you provide some more details on the framework analysis approach and on how you retrieved the themes at pre-intervention and 3-months follow-up?
P. 13: In my view, you might discuss the depressive symptoms of the mothers as another potential limitation of your study. Which results would you have expected if mothers were in fact diagnosed with a full-blown diagnosis?
Round 2
Reviewer 1 Report
1. Background:
The background section is structured and provides a brief, but comprehensive overview of the current research available, a clearly defined definition and the contextualization of intellectual disability and the burden such a diagnosis has on society and the family itself. The author then provides more information on the impact on maternal mental health, and current interventions were discussed. Positive parenting studies as well as the routine care in low-income countries were discussed. The intervention LTP-IMOW and the research project's aims are clearly defined. The following suggestions should be taken into consideration:
2. Method:
The methods section consists of 9 subsections. The following was of note within each section:
· Design and Settings:
o The research design was mentioned. Both the qualitative and quantitative methods were identified. Furthermore, the ethical considerations were adequately discussed. A brief contextualization of the target group was discussed.
· Recruitment and Randomization
o The recruitment process, as well as the inclusion and exclusion criteria, was clearly defined. Ethical considerations were made. The authors also discussed the randomization process sufficiently.
o Editing mistake
§ (Community mobilizers are members 140 from within a particular community who can bring together as many stakeholders)
· The intervention: LTP in my own way plus
o The intervention aims and the theory it is grounded in are discussed. The two components of this intervention are briefly mentioned. The sequencing of the intervention and the training of the trainers was highlighted.
· Routine care
o The usual care is discussed and the author ellaborated on the resources availble within the usual care setting. The authors also ensured amd kept track of this group’s mental healthcare needs.
· Feasibility of the study procedures & Acceptability of the intervention
o Both these sections were adequately discussed.
· Qualitative evaluation
o The authors restructered this section and provided more information about the development and the implementation of the various topic guides. The qualitative method was also described and language barriers were taken into account
· Exploratory evaluation of outcomes
o Maternal depression and child outcomes were discussed, and assessments were also mentioned. Measurement moments were also mentioned:
· Qualitative Analysis and Satistical Analysis
o Both these sections were adequately discussed.
3. Results:
Qualitative section:
The Qualitative analysis begins with a breakdown of the themes that were identified during the pre-intervention phases and post-intervention phases. These themes reflect the content of Table 1. Furthermore, the authors referred to the table throughout the section which allowed for a cohesive text.
Quantitative section:
This section is clear and concise, with relevant results. The participant characteristics were discussed briefly, and the author discussed the feasibility evaluation of the study. The quantitative outcomes were discussed, and the restructing of Table 2 is informative and provides enough information to the reader.
4. Discussion:
The discussion section integrated and briefly summarised this study's aims and quantitative and qualitative results. It also emphasised the importance of this study within developing countries. It was noted that some of the text was duplicated, this should be revised (e.g . The results of this feasibility study show excellent recruitment and retention of moth- 641 ers who have screened positive for depression in this study and the high uptake of the 642 LTP-IMOW plus intervention…..). Furthermore, the authors linked the results back to the greater body of knowledge within this field of research. Future research recommendations were made and limitations were discussed.
5. Conclusion:
The conclusion briefly studies the research project and the results obtained.
Reviewer 4 Report
This is clearly a very improved version of the submitted paper and the authors are to be applauded for their thorough answers and adaptations. There is one minor comment I would like to raise:
Table 2: Did you check for multiple comparisons and respective adjustments of the alpha level?
Once this point has been clarified, the paper is ready for publication, in my view.
